# Regularizing Deep Multi-Task Networks using Orthogonal Gradients

## Abstract

Deep neural networks are a promising approach towards multi-task learning because of their capability to leverage knowledge across domains and learn general purpose representations. Nevertheless, they can fail to live up to these promises as tasks often compete for a model's limited resources, potentially leading to lower overall performance. In this work we tackle the issue of interfering tasks through a comprehensive analysis of their training, derived from looking at the interaction between gradients within their shared parameters. Our empirical results show that well-performing models have low variance in the angles between task gradients and that popular regularization methods implicitly reduce this measure. Based on this observation, we propose a novel gradient regularization term that minimizes task interference by enforcing near orthogonal gradients. Updating the shared parameters using this property encourages task specific decoders to optimize different parts of the feature extractor, thus reducing competition. We evaluate our method with classification and regression tasks on the multiDigitMNIST, NYUv2 and SUN RGB-D datasets where we obtain competitive results.

## 1 Introduction

Deep neural networks have proven to be very successful at solving isolated tasks in a variety of fields ranging from computer vision to NLP. In contrast to this single task setup, multi-task learning aims to train one model on several problems simultaneously. This approach would incentivize it to transfer knowledge between tasks and obtain multi-purpose representations that are less likely to overfit to an individual problem. Apart from potentially achieving better overall performance (Caruana, 1997), using a multi-task approach offers the additional benefit of being more efficient in memory usage and inference speed than training several single-task models (Teichmann et al., 2018).

A popular design for deep multi-task networks involves hard parameter sharing (Ruder, 2017), where a model contains a common encoder, which is shared across all tasks and several problem specific decoders. Given a single input each of the decoders is then trained for a distinct task using a different objective function and evaluation metric. This approach allows the network to learn multi-purpose representations through the shared encoder which every decoder will then use differently according to the requirements of its task. Although this architecture has been successfully applied to multi-task learning (Kendall et al., 2018; Chen et al., 2017) it also faces some challenges. From an architectural point of view it is unclear how to choose the task specific network capacity (Vandenhende et al., 2019; Misra et al., 2016) as well as the complexity of representations to share between tasks. Additionally, optimizing multiple objectives simultaneously introduces difficulties based on the nature of those tasks and the way their gradients interact with each other (Sener & Koltun, 2018). The dissimilarity between tasks could cause negative transfer of knowledge (Long et al., 2017; Zhao et al., 2018; Zamir et al., 2018) or having task losses of different magnitudes might bias the network in favor of a subset of tasks (Chen et al., 2017; Kendall et al., 2018). It becomes clear that the overall success of multi-task learning is reliant on managing the interaction between tasks, and implicitly their gradients with respect to the shared parameters of the model.

This work focuses on the second category of challenges facing networks that employ hard parameter sharing, namely the interaction between tasks when being jointly optimized. We concentrate on reducing task interference by regularizing the angle between gradients rather than their magni-

tudes. Based on our empirical findings unregularized multi-task networks have high variation in the angles between task gradients, meaning gradients frequently point in similar or opposite directions. Additionally, well-performing models share the property that their distribution of cosines between task gradients is zero-centered and low in variance. Nearly orthogonal gradients will reduce task competition as individual task decoders learn to use different features of the encoder, thus not interfering with each other. Furthermore, we discover that popular regularization methods such as Dropout (Srivastava et al., 2014) and Batchnorm (Ioffe & Szegedy, 2015) implicitly orthogonalize the task gradients. We propose a new gradient regularization term to the multi-task objective that explicitly minimizes the squared cosine between task gradients and show that our method obtains competitive results on the NYUv2 (Nathan Silberman & Fergus, 2012) and SUN RGB-D (Song et al., 2015) datasets.

## 2   RELATED WORK

Multi-task learning is a sub-field of transfer learning (Pan & Yang, 2009) and encompasses a variety of methods (Caruana, 1997). The recent focus on deep multi-task learning can be attributed to the neural network's unparalleled success in computer vision (Krizhevsky et al., 2012; Simonyan & Zisserman, 2014; He et al., 2016) and its capability to create hierarchical, multi-purpose representations (Bengio et al., 2013; Yosinski et al., 2014). Deep multi-task learning is commonly divided into hard or soft parameter sharing methods (Caruana, 1997; Ruder, 2017). Soft parameter sharing maintains separate models for each task but enforces constraints on the joint parameter set (Yang & Hospedales, 2016). In this work we focus solely on hard parameter sharing methods, which maintain a common encoder for all tasks but also contain task-specific decoders that use the learned generic representations.

We further split deep multi-task approaches into architecture and loss focused methods. Architecture based methods aim at finding a network structure that allows optimal knowledge sharing between tasks by balancing the capacities of the shared encoder and the task specific decoders. Most multi-task related work chooses the architecture on an *ad hoc* basis (Teichmann et al., 2018; Neven et al., 2017), but recent research looks to answer the question of how much and where to optimally share knowledge. Cross-stitch networks maintain separate models for all tasks but allow communication between arbitrary layers through specialized cross-stitch units (Misra et al., 2016). Branched multi-task networks allow for the decoders to also be shared by computing a task affinity matrix that indicates the usefulness of features at arbitrary depths and for different problems (Vandenhende et al., 2019). Liu et al. (2019b) introduces attention modules allowing task specific networks to learn which features from the shared feature network to use at distinct layers.

Loss focused methods try to balance the impact of individual tasks on the training of the network by adaptively weighting the task specific losses and gradients. Certain tasks might have a disproportionate impact on the joint objective function forcing the shared encoder to be optimized entirely for a subset of problems, effectively starving other tasks of resources. Kendall et al. (2018) devise a weighting method dependent on the homoscedastic uncertainty inherently linked to each task while Chen et al. (2017) reduce the task imbalances by weighting task losses such that their gradients are similar in magnitude. Sener & Koltun (2018) cast multi-task learning as a multi-objective optimization problem and aim to find a Pareto optimal solution. They also analyze gradients but with the goal to then scale these such that their convex combination will satisfy the necessary conditions to reach the desired solution. In contrast to these approaches our method does not seek to scale gradients, neither directly nor via task weights, but conditions the optimization trajectory towards solutions that have orthogonal task gradients.

The problem of conflicting gradients or task interference has been previously explored in multi-task learning as well as continual learning. Zhao et al. (2018) introduce a modulation module that reduces destructive gradient interference between tasks that are unrelated. Du et al. (2018) choose to ignore the gradients of auxiliary tasks if they are not sharing a similar direction with the main task. Riemer et al. (2018) maximize the dot product between task gradients in order to overcome catastrophic forgetting. These methods have in common the interpretation that two tasks are in conflict if the cosine between their gradients is negative, while alignment should be encouraged. Our work differs from this perspective by additionally penalizing task gradients that have a similar direction, arguing that by decorrelating updates the shared encoder is able to maximize its representational capacity. A

similar observation regarding orthogonal parameters is made by Rodríguez et al. (2016) who propose a weight regularization term for single task learning that decorrelates filters in convolutional neural networks. Another way to minimize interference is to encourage sparsity (French, 1991; Javed & White, 2019), although doing so directly might interfere with the network's ability to learn shared representations (Ruder, 2017) .

Finally our work is in line with recent research (Liu et al., 2019a; Santurkar et al., 2018) that emphasizes the benefit of analyzing gradients to understand neural networks and devise potential improvements to their training. We share elements with Drucker & Le Cun (1991) and more recently Varga et al. (2017) in that we propose explicit regularization methods for gradients.

## 3  ORTHOGONAL TASK GRADIENTS

In this work we present a novel gradient based regularization term that orthogonalizes the interaction between multiple tasks. We define a multi-task neural network as a shared encoder $f_{\theta_{sh}}$ and a set of task-specific decoders $f_{\theta_{t_i}}$, for each of the $T$ tasks $\mathcal{T} = \{t_1, ..., t_T\}$. The encoder creates a mapping between the input space $\mathcal{X}$ and a latent feature space $\mathbb{R}^d$ that is used by each of the decoders to predict the task specific labels $\mathcal{Y}^{t_i}$. Each of the inputs in $\mathcal{X}$ is associated to a set of labels for the tasks in $\mathcal{T}$, forming the dataset $\mathcal{D} = \{x_i, y_i^{t_1}, ..., y_i^{t_T}\}_{i \in N}$ of $N$ observations.

For task $t \in \mathcal{T}$ we define the empirical loss as $\mathcal{L}_t \triangleq \frac{1}{N} \sum_{i \in N} \mathcal{L}_t(f_{\theta_t}(f_{\theta_{sh}}(x_i)), y_i)$. The multi-task objective can be then constructed as a convex combination of individual task losses using the weights $w_t \in \mathbb{R}$:

$$\mathcal{L}_{\mathcal{T}} = \sum_{t \in \mathcal{T}} w_t \mathcal{L}_t \tag{1}$$

Using gradient descent to minimize the multi-task loss in Equation 1, we obtain the following update rule for the parameters $\theta_{sh}$:

$$\theta_{sh} = \theta_{sh} - \gamma \sum_{t \in T} w_t \frac{\partial \mathcal{L}_t}{\partial \theta_{sh}} \tag{2}$$

It becomes clear that the overall success of a multi-task network is dependent on the individual task gradients and their relationship to each other. Task gradients might cancel each other out or a certain task might dominate the direction of the encoder's parameters. We further examine the interaction between two tasks $t_i$ and $t_j$ by looking at the cosine of their gradients with respect to the encoder:

$$\cos(t_i, t_j) = \cos(\frac{\partial \mathcal{L}_{t_i}}{\partial \theta_{sh}}, \frac{\partial \mathcal{L}_{t_j}}{\partial \theta_{sh}}) \tag{3}$$

Previous work argues that negative transfer, task interference or competition (Du et al., 2018; Sener & Koltun, 2018; Zhao et al., 2018) happens when this cosine is negative, leading to tasks with smaller gradient magnitudes in fact increasing their error during training. The interference between tasks lies in the competition for resources in the shared encoder $f_{\theta_{sh}}$. Based on empirical observations we argue that multi-task networks not only benefit when the cosine is non-negative but more so when task gradients are close to orthogonal. In a continual learning setting it makes sense for task gradients to be as aligned as possible in order to avoid catastrophic forgetting (Riemer et al., 2018). In a multi-task setting it is however unclear whether maximizing the transfer between tasks also leads to a superior solution for all objectives, especially since there is no risk of forgetting. In our experiments we observe a higher performance when orthogonalizing correlated tasks on the SUN RGB-D dataset. Standley et al. (2019) make a similar finding that learning related tasks does not necessarily improve the multi-task optimization.

In our approach we minimize the squared cosine during training which diminishes competition as each task will be able to optimize different parameters of the encoder. By also orthogonalizing positive transfer the encoder will produce a richer feature space and multi-purpose representations.

To minimize the cosine between two task gradients we simply add the squared cosine to the multi-task objective function from Equation 1 with an additional hyper-parameter $\alpha \in \mathbb{R}$ to adjust the penalty weight:

$$\mathcal{L}_{t_i t_j} = w_{t_i} \mathcal{L}_{t_i} + w_{t_j} \mathcal{L}_{t_j} + \alpha \cos^2(t_i, t_j) \tag{4}$$

We can generalize Equation 4 to $T$ tasks by taking the squared Frobenius norm of the cosine distance matrix between gradients. We define $\nabla_{\theta_{sh}}$ as the column vector of unit normalized partial derivatives of the task losses with respect to $\theta_{sh}$. The distance matrix for $T$ tasks $(\cos^2(t_i, t_j))_{1 \le i,j \le T}$ can be then efficiently computed by taking the outer product of $\nabla_{\theta_{sh}}$ with itself. Subsequently we subtract the constant distance between identical tasks and normalize by accounting for the matrix symmetry as well as the number of task pairs in order to ensure the same bounds for the penalty term.

$$\nabla_{\theta_{sh}} = \left( \frac{\partial \hat{\mathcal{L}}_{t_1}}{\partial \theta_{sh}}, \frac{\partial \hat{\mathcal{L}}_{t_2}}{\partial \theta_{sh}}, ..., \frac{\partial \hat{\mathcal{L}}_{t_T}}{\partial \theta_{sh}} \right)$$

$$\mathcal{L}_{\mathcal{T}} = \sum_{t \in \mathcal{T}} w_t \mathcal{L}_t + \frac{\alpha}{T(T-1)} \| \nabla_{\theta_{sh}}^{\intercal} \nabla_{\theta_{sh}} - I_T \|_F^2 \tag{5}$$

The above equation generalizes the gradient regularization term for $T$ tasks and maintains its range within $[0, 1]$. Even though $w_t$ are hyper-parameters, our focus is solely on the cosine regularization and will therefore treat them as constants. In all the experiments we add our penalty term to the naive approach of having all tasks equally weighted. Computing the regularization term for each layer in the shared encoder is computationally prohibitive, so in practice we restrict ourselves to computing the loss with respect to only the last layer of the encoder.

Finally, we will refer to $\Phi_{(t_i, t_j)}$ as the distribution of cosines between the gradients of $t_i$ and $t_j$ throughout training, having mean $\mu_{(t_i, t_j)}$ and standard deviation $\sigma_{(t_i, t_j)}$. Our gradient regularization method minimizes $\sigma_{(t_i, t_j)}$, which will be empirically shown later on.

## 4 EMPIRICAL ANALYSIS

To illustrate our findings we will use the MultiDigitMNIST dataset (Sun, 2019) with a minor alteration to make it more suitable for multi-task learning. MultiDigitMNIST is a dataset constructed by positioning two MNIST digits side by side on an image of 64 by 64 pixels. Each digit is located at an arbitrary location within its half, varying in style and orientation as in the original MNIST dataset. The resulting tasks are classifying the left and right digits in the image, denoted $t_{left}$ and $t_{right}$ respectively. We modify the setup by choosing a subset of possible digits for each task, creating two disjoint sets of labels. This allows the tasks to be related, as both classify digits, but at the same time avoiding redundancy since each task is optimized on different labels. By making this modification we encourage each decoder to learn task specific features, while still taking advantage of the shared filters trained for generic digit classification. For our experiments we have assigned even digits to $t_{left}$ and odd digits to $t_{right}$, as shown in Figure 1. The resulting dataset contains 16000 images for training, 4000 for validation and 5000 for testing. It is worth noting that even though the combination of left and right digits are random, it is ensured that individual pairs of digit classes are only present in one dataset split. This guarantees that the network is evaluated on new digit pairings rather than pairings seen during training.

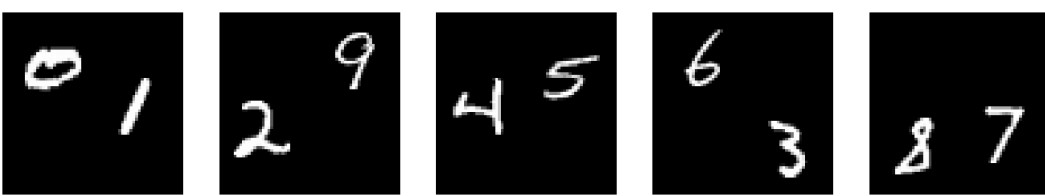

Figure 1: Sample images from the modified multiDigitMNIST dataset (Sun, 2019). Only even digits are assigned to $t_{left}$ while $t_{right}$ contains odd numbers. The same combination of digits in an image does not appear in multiple dataset splits.

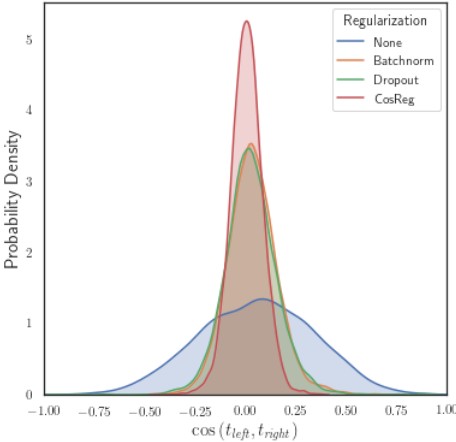
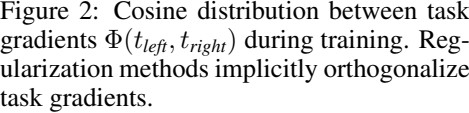
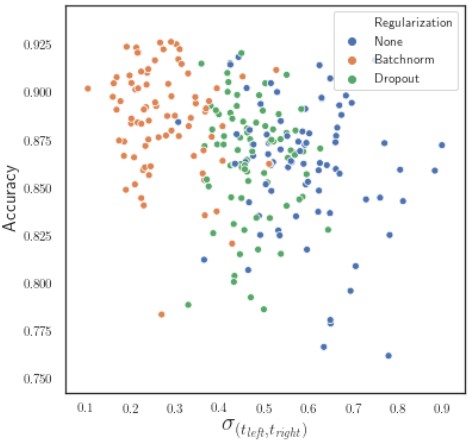

Figure 2: Cosine distribution between task gradients $\Phi(t_{left}, t_{right})$ during training. Regularization methods implicitly orthogonalize task gradients.

Figure 3: Evaluation of models with different hyper-parameters and random seeds. The final validation accuracy is plotted against the standard deviation of gradient cosines from the first training epoch.

We perform our experiments using a convolutional neural network architecture. The shared encoder consists of two convolutional layers, while the decoders have one convolutional and two fully connected layers. The decoders contain convolutions in order to encourage the learning of task-specific filters. For simplicity our convolutional layers lack bias terms and use stride to replace maxpooling layers. We have tested the configuration with bias terms and maxpooling layers and do not encounter a noticeable difference. Training is performed using the cross-entropy loss and the Adam (Kingma & Ba, 2014) optimizer. To evaluate the overall performance of a model we measure the harmonic mean of the accuracy it obtains for both tasks on the validation set.

To derive an accurate picture of the interaction between task gradients we perform our analysis on a variety of instances using a range of hyper-parameters and random seeds. We vary the network capacity by having different number of filters in the convolutional layers, thus allowing different ratios of resources between encoder and decoders. The training is being varied by iterating over different batch sizes and learning rates. Each unique configuration is being evaluated on five different initializations.

We compare our method with both regularized and unregularized baselines and show the cosine distributions $\Phi_{(t_i, t_j)}$ during a sample training in Figure 2. It can be observed that the unregularized model displays a distribution with high variance, where gradients frequently form sharp and obtuse angles. Intuitively having gradients in opposite directions will hurt performance, while having them in the same direction raises questions about the usefulness of optimizing multiple objectives. On the other hand Dropout and Batchnorm seem to implicitly reduce the variance of these angles, favoring orthogonality between task gradients. This confirms the findings of Santurkar et al. (2018) that Batchnorm is having a smoothing effect on the loss surface and extends it to multi-task scenarios. Unsurprisingly, the regularized models outperform the unregularized baseline as seen in Table 1. This leads to the question whether a model explicitly regularized for gradient orthogonality can help the training of multi-task networks. Similar to the findings of Liu et al. (2019a) and Santurkar et al. (2018) we observe in our analysis that high gradient variance is more prominent in the beginning of training and when using smaller batch sizes. We find that unregularized models also reduce their cosine variance in later stages of training as they reach convergence and further explore this in Appendix A. Figure 3 shows the validation accuracy and standard deviation $\sigma_{(t_i, t_j)}$ during the first training epoch of several models. We notice that this initial standard deviation is a good indicator of the final generalization performance of a model. Additionally we observe a clear separation in terms of $\sigma_{(t_i, t_j)}$ between regularization methods. We believe that by having non interfering gradients in the beginning of training, the model is being guided into different regions of the search space that ultimately prove to yield better local minima.

Based on these observations we evaluate our gradient regularization method and observe that $\sigma_{(t_i,t_j)}$ is being successfully reduced as seen in Figures 2 and 5. The evaluated model contains 20 filters on each convolutional layer, and has been optimized using a learning rate of $10^{-3}$ with batches containing 64 images. The final test scores over five runs are shown in Table 1. Although our method does not on seem to have a major impact by itself, it significantly boosts the performance of the model when used in conjunction with Batchnorm. We believe the added complexity of the loss landscape is benefiting from the smoothing effect of Batchnorm (Santurkar et al., 2018), which is why the methods work well together. Although Batchnorm has already a gradient orthogonalizing effect on training, further regularizing them proves to be beneficial. It is worth noting that in these experiments none of the multi-task approaches outperform their single task counterparts. We believe this is due to the abundance of data compared with the relative simplicity of the tasks. This stands in contrast to the results on the following benchmark datasets which have much less data compared with the amount of variability displayed.

Table 1: Harmonic mean of task accuracies with standard deviation, as well as the standard deviation of the cosine distribution on the modified MultiDigitMNIST dataset. Regularization methods implicitly reduce $\sigma_{(t_i,t_j)}$.

| **Model** | Acc. (%) | $\sigma_{(t_\text{left},t_\text{right})}$ |
|---|---|---|
| Single Task | **93.7 (0.00)** | - |
| No reg | 90.7 (0.01) | 0.27 |
| Dropout | 91.0 (0.00) | 0.11 |
| Batchnorm | 91.3 (0.02) | 0.11 |
| CosReg($\alpha = 10$) | 90.7 (0.00) | 0.03 |
| CosReg($\alpha = 0.1$) + Batchnorm | **92.5 (0.01)** | 0.09 |

## 5 BENCHMARKS

For our experiments we use the multi-task friendly SegNet (Badrinarayanan et al., 2017) architecture. The model consists of symmetric VGG16 (Simonyan & Zisserman, 2014) encoder and decoders. The decoders perform upsampling using the indices obtained from the maxpooling layers in the encoder. Due to limited number of data points the network is being initialized with the weights from a VGG16 network pre-trained on ImageNet. We use independent decorders for each task in order to evaluate the gradients on the last layer of the VGG encoder. All experiments have been implemented in PyTorch and run on a TITAN X PASCAL GPU machine. We compare the performance of our approach with GradNorm (Chen et al., 2017) and Kendall et al. (2018) as both methods are focused on the interaction between tasks rather than architecture design. It is worth reiterating that our method differentiates itself from these approaches by not scaling task gradients, hence in all experiments the task weights for CosReg are set to 1.

### 5.1 NYUv2

We evaluate our regularization method on the NYUv2 dataset (Nathan Silberman & Fergus, 2012) of indoor scenes. The small dataset of 795 training and 654 test images includes image segmentation, depth and surface normal labels which makes it a suitable benchmark having both classification and regression tasks. The dataset is challenging as it contains indoor images from multiple viewpoints and under different lighting conditions displaying high variation relative to the number of data points it offers. We do not augment the dataset with auxiliary observations that have only a subset of labels, such as additional video frames. The original input images of $640{\times}480$ pixels are resized to $320{\times}320$, while the target images are downsampled to $80{\times}80$. This allows the model to have less memory requirements while still handling images with semantic significance.

**Image segmentation.** Each pixel in the target image for the segmentation task is labeled as one of 14 classes (bed, chair, window etc.) including the background. We train the task to minimize the pixel-wise cross-entropy loss, while using the mean intersection over union (IoU) metric to evaluate it.

Table 2: Task errors for the NYUv2 dataset. Lower values are preferred and the best performance for each task is displayed in bold.

| Model | Segmentation $1-$ mIoU | Depth Estimation RMSE | Surface Normal $1 - |\cos|$ |
|---|---|---|---|
| Single Task | 0.663 | 0.775 | **0.051** |
| Equal Task Weights | 0.670 | 0.765 | 0.053 |
| Gradnorm (Chen et al., 2017) | 0.651 | 0.747 | 0.052 |
| Kendall et al. (2018) | 0.659 | **0.745** | 0.052 |
| CosReg | **0.646** | 0.747 | 0.053 |

**Depth estimation.** The indoor images were captured with a Microsoft Kinect which can collect depth information. Each pixel of the target image for this task is annotated with the distance in meters. We use mean squared error (MSE) loss to optimize the task decoder and evaluate it using the root MSE metric.

**Surface normals.** The surface normals for each image were generated algorithmically and are encoded over three channels representing each axis. Predictions are normalized to unit length and trained to minimize the MSE loss. A model's performance is evaluated by computing the cosine between the target and predicted surface normals at each pixel.

All models are trained using Adam for 25 epochs, with a static learning rate of $10^{-4}$. Due to memory constraints resulting from having three decoders we use a batch size of 2. During training the variation in loss amplitude is large do to the optimization of three objective functions. In order for our gradient regularization method to better adapt to these changes we also try scheduling $\alpha$ to be multiplied by the average loss of the tasks. The weight has the sole purpose of scaling the regularization loss term and is not being backpropagated through. For these experiments we use an $\alpha$ of 10.

The results are shown in Table 2. We also evaluate the single task approach, where one model is being trained to optimize only one objective at a time, not receiving any signal from the other tasks. It can be seen that adopting a naive multi-task approach of assigning equal weights to each objective produces mixed results. While depth estimation benefits from the multi-task setting the performance for semantic segmentation and surface normal prediction is reduced. Similar to Kendall et al. (2018) and Chen et al. (2017) our method also improves on the performance of the naive multi-task approach. It is worth emphasizing that both Chen et al. (2017) and Kendall et al. (2018) explicitly weigh individual losses to balance the training between tasks. In contrast, our method operates solely on regularizing the direction of gradients and not their magnitude, while achieving similar results.

## 5.2 SUN RGB-D

We further test our proposed method on the SUN RGB-D dataset (Song et al., 2015). Similar to the NYUv2 dataset it contains images describing indoor scenes but with 5285 train and 5050 test images it is significantly larger. The tasks the models are trained to solve are also semantic segmentation and depth estimation, but for the former we have both fine and coarse labels at our disposal. This allows us to construct two similar tasks with different difficulties as the coarse segmentation offers 13 classes while for the fine segmentation we are provided with 37. Having such a strong similarity between tasks will bias the mode of the cosine distribution since optimizing one task will inevitably benefit the other. The fact that the tasks have predominantly sharp angles between gradients can be seen in Figure 4. We can observe that even in this situation our method successfully produces orthogonal gradients. This setup is interesting as it allows us to test our hypothesis of the benefits of gradient orthogonality even when tasks are not in competition with each other.

We train the models using Adam for 25 epochs using a batch size of 3 images and exponentially decay the learning rate from a base of $10^{-4}$ by 0.5 every 2 epochs. Similar to the previous setup we resize the input images to 320×320 and the outputs to 80×80, but additionally augment this with horizontal flips. For these experiments we also found the model to work best with an $\alpha$ of 10.

Table 3: Task errors for the SUN RGB-D dataset. The segmentation tasks are correlated as they are performed at coarse (13 classes) and fine (37 classes) granularities.

| Model | Segmentation (13) $1-$ mIoU | Segmentation (37) $1-$ mIoU | Depth Estimation RMSE |
|---|---|---|---|
| Single Task | 0.649 | 0.719 | 0.586 |
| Equal Task Weights | 0.646 | 0.717 | 0.586 |
| Gradnorm (Chen et al., 2017) | 0.648 | 0.723 | 0.606 |
| Kendall et al. (2018) | 0.646 | 0.716 | **0.582** |
| CosReg | **0.644** | **0.714** | 0.584 |

The results of the benchmark are displayed in Table 3. Similar to NYUv2 the single task baselines are outperformed by most multi-task approaches. Although the dataset is larger we believe each task still does not have sufficient observations to be indifferent to knowledge transfer. As can be observed our method outperforms all other baselines and we believe this is due to the orthogonalization of updates from the correlated segmentation tasks. This suggests that tasks that are related under transfer learning do not necessarily help each other during multi-task. A similar observation has been made by Standley et al. (2019) who obtain superior multi-task results by combining uncorrelated tasks.

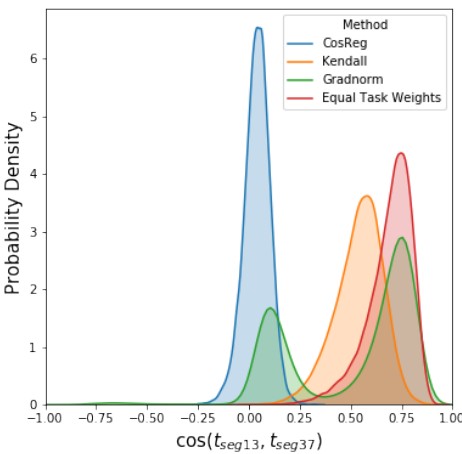

Figure 4: Cosine distribution between correlated segmentation tasks. Even though the gradients are in a highly dimensional space they point in similar directions throughout training. CosReg successfully regularizes even in this circumstance.

## 6 CONCLUSION

In this work we explore the interaction between task gradients during training. Through an empirical analysis on the multiDigitMNIST dataset we observe that unregularized models have high variance in the angles between task gradients, while models with lower variance tend to perform better. Additionally we find that common regularization methods such as Dropout and Batchnorm implicitly orthogonalize gradients throughout training, thus minimizing task interference. Based on this finding we propose a novel gradient regularization term that explicitly orthogonalizes task gradients and obtain competitive results on the NYUv2 and SUN RGB-D datasets. Different to recent approaches, this method balances tasks by regularizing the direction of gradients rather than their magnitude.

In future work we would like to further explore the impact of gradient regularization at the beginning of training and measure its effects on later stages of the optimization. In our experiments on the multiDigitMNIST dataset we found a correlation between the initial cosine variance and final validation score. This topic should be further analyzed to evaluate the predictive power of this measure and if it can alleviate the need for a validation set. Moreover, we would like to further improve our gradient regularizer by investigating methods to dynamically scale the loss term. Even though it provides simplicity, under the current formulation the cosine loss has an upper bound independent of tasks, relying on the manual adjustment of $\alpha$ for each domain.

In line with recent research we believe there is a lot to be gained by analyzing and influencing gradients throughout training. This proves to be especially true in multi-task learning where managing the interaction between tasks plays a crucial role on the success of a model.

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

## A    COSINE STD DURING TRAINING

The decrease of $\sigma_{(t_i,t_j)}$ during training on MultiDigitMNIST can be seen in Figure 5. As opposed to unregularized models, networks using Dropout and Batchnorm start training with reduced cosine variance and maintain the values stable after the first epochs. The vanilla network also decreases the standard deviation of its cosine distribution during training but at a far slower rate.

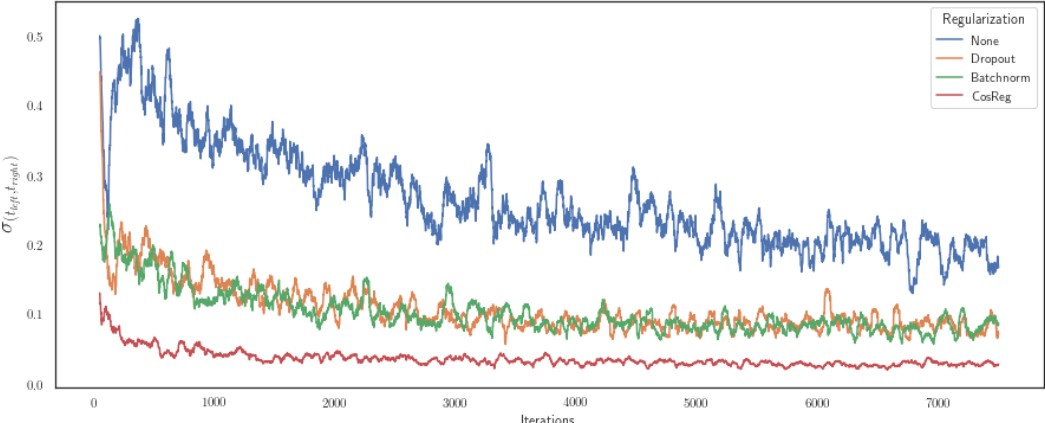

Figure 5: Moving standard deviation of $\cos(t_{\text{left}}, t_{\text{right}})$ throughout training on MultiDigitMNIST. The standard deviation is computed over rolling windows of 50 iterations. It can be observed that even for the vanilla model $\sigma_{(t_{\text{left}}, t_{\text{right}})}$ decreases as training progresses, but remains at relatively high values compared to the regularized networks.

