# OpenReview forum: "Regularizing Deep Multi-Task Networks using Orthogonal Gradients"
_ICLR.cc/2020/Conference — Reject_

### Official Review · AnonReviewer2 · 2019-10-22
**Official Blind Review #2**

**Rating:** 3

**Review:**

Summary: In this paper, the author analyzed gradient regularization in deep multitask learning. They empirically discovered a sharper concentration (low variance) in angles between the task gradient distributions could potentially improve the performance in multi-task learning. Then they proposed a new gradient regularization to enforce the gradient for each task be orthogonal. Empirical results (on Multi-digits MNIST and NYUv2 data-sets) indicate a marginal improvement, comparing with baselines.

Main Comments:

The discovering in the paper sounds interesting while the work looks like preliminary and unpolished. Particularly I have the following technical and conceptual concerns:
1.  In high dimensional geometrical space, ** the random high dimensional vectors will be orthogonal with high probability**. The authors can find this conclusion in many places, for example:

https://courses.cs.washington.edu/courses/cse521/16sp/521-lecture-6.pdf
(Theorem 6.3)
https://www.cs.princeton.edu/courses/archive/fall14/cos521/lecnotes/lec11.pdf
(Corollary 2)

I noticed the author claimed in the paper “Based on empirical observations presented later on we argue that multi-task networks not only benefit when the cosine is non-negative but more so when task gradients are close to orthogonal.” (Page 3).  However, this empirical claim is not convincing for me.  Since gradient of the over-parameterized neural network is in very high dimension, the task gradients closing to orthogonal may happen because of
(a) the internal high-dimensional geometry property
(b) or the phenomene caused by the deep multi-task learning

It will be much better and more clear if the author can provide more analytical, theoretical (e.g provable bound even on linear model) or empirical (e.g ablation study) support to understand the behaviors of gradient in high dimensional problem.

2. In the proposed approach, I am not clear how the author choose the **task weights** (e.g in Eq.1) “w_{t_i}” ? It seems that the author set them as hyper-parameters. Since understanding the task relations and automatically estimating their relationships is the key factor for avoiding negative transfer in multitask learning (e.g Sener [2018]), I think it is not a proper way to simply adjust them as hyper-parameters.

3. In the experimental part, the proposed approach showed almost no or very marginal improved performance. The author should provide more evidences to show the utility or potentials, in order to convince the community to adopt the proposed approach.

Minor comments:
1. From equation (4) to (5) is not obvious, it will be better to provide the derivation details.
2. The analyzed problem sounds more like the “multi-output” or “multi-label” problem (Section 3 in the paper). In the multi-task learning generally the inputs for each task X are different.

Overall, I feel it is an interesting and promising direction to consider gradient regularization based approach in multi-task learning. However, the current manuscript is not mature for the acceptance.


Reference:
Multi-Task Learning as Multi-Objective Optimization.  Ozan Sener, Vladlen Koltun, NeurIPS, 2018
----------------------------------------------------------------------------------------------------

After rebuttal

I thank the author for detailed rebuttal and efforts for making the paper better.

I accept points  (1), (2) and (5) , then I update my score to weak reject -- 3.

The main reason that I still keep the decision toward rejection is the experimental part.

Since it is an empirical paper, I suggest the author either systematically show more comparisons and more datasets.
Or the author can develop some theoretical insights/bounds (e.g point (1)) for enhancing the contribution of the paper.





**Experience Assessment:**

I have published one or two papers in this area.

**Review Assessment: Checking Correctness Of Derivations And Theory:**

N/A

**Review Assessment: Checking Correctness Of Experiments:**

I assessed the sensibility of the experiments.

**Review Assessment: Thoroughness In Paper Reading:**

I read the paper at least twice and used my best judgement in assessing the paper.

---

> ### Author Response · Authors · 2019-11-14
> **Response**
>
> We thank the reviewer for the feedback and the provided references.
>
>
> 1. Orthogonal gradients in high dimensional spaces
> We agree with the reviewer that due to the curse of dimensionality any pair of random vectors will be nearly orthogonal in high dimensional spaces. Nevertheless, we would like to point out that even though deep neural networks are heavily over-parametrized they also have highly correlated features, redundancies and dead neurons. In practice this often leads to representations and gradients that form sharp or obtuse angles. We have added an additional figure which shows the cosine distribution between gradients of related tasks. We use the SUN RGB-D dataset which contains two segmentation tasks, one coarse and another one that is fine grained. The tasks are very similar as pixels are often attributed to the same class in both tasks. We can observe that throughout training the two task gradients point in similar directions, even though we are talking about a space with over 2 million dimensions.
>
> 2. Task weights as hyper-parameters
> We realize that we didn't emphasize that our method does not require setting task weights. This is one aspect on how we differentiate ourselves from other MTL approaches, since our method is solely focused on regularizing gradient angles and does not seek to alter their magnitude. For all our experiments the task weights $w_t$ are set to 1 and we have updated our paper to explicitly state so.
>
> 3. Performance
> We have provided benchmarks on an additional dataset (SUN RGB-D), where we obtain improved results.
>
> 4. Generalization of CosReg
> We have updated our paper to include more details about the generalization of CosReg to T tasks in equation 5.
>
> 5. Multi-task vs multi-output
> From the point of view of the used datasets our problem setting is indeed related to multi-output. This however does not contradict the multi-task paradigm, as each output belongs to a different task which has its own conditional distribution, objective function, complexity and evaluation metric. We do acknowledge the similarity, especially if multi-output is viewed as a subset of multi-task learning. We would also like to add that our method is not restricted to this setup and can potentially be used for tasks with different inputs.

---

### Official Review · AnonReviewer1 · 2019-10-23
**Official Blind Review #1**

**Rating:** 3

**Review:**

This paper embraces the idea that better multi-task/lifelong learning can be achieved if tasks produce gradients that are orthogonal to the gradients produced by other tasks. The authors propose an approach to regularizing learning in order to incentivize this to happen. However, as they mention themselves, the regularized loss is computationally intractable in general and they only apply it to a subset of their network as a result. Given the computational scalability concerns, it is natural to wonder why researchers in the community would adopt this approach rather than other approaches that also aim to make gradients orthogonal.

The idea of producing orthogonal gradients across tasks or examples is not new in the context of lifelong/multi-task learning. In fact, just to name a few, [1] demonstrated that noise alone can lead to orthogonal gradients, [2] demonstrated that modular neural network architectures can lead to orthogonal gradients and less interference. Additionally, sparsity naturally leads to orthogonal gradients as does the recent approach in [3].  These approaches achieve orthogonal gradients without adding a significant computational burden to learning. This paper can be greatly improved by discussing past approaches to producing orthogonal gradients and why they are theoretically / empirically worse than CosReg.

[1] "A theory for how sensorimotor skills are learned and retained in noisy and nonstationary neural circuits". Robert Ajemian, Alessandro D’Ausilio,  Helene Moorman, and  Emilio Bizzi. PNAS'13.

[2] "Routing Networks: Adaptive Selection of Non-linear Functions for Multi-Task Learning". Clemens Rosenbaum, Tim Klinger, and Matthew Riemer. ICLR'18.

[3] "Meta-Learning Representations for Continual Learning". Khurram Javed and Martha White. 2019.

Additionally, despite much past work, I tend to think that the entire quest for orthogonal gradients is not particularly well motivated as it is missing half of the story. Orthogonal gradients only address the problem of interference during learning, but don't help maximize transfer during learning. In fact, intuitively Figure 2 showcases that CosReg diminishes transfer during learning in comparison to baselines. Some recent work, such as [4] and [5], argues that what we really want is to maximize the dot product of gradients i.e. their alignment. This perspective achieves the best of both worlds as it incentivizes orthogonality to address interference while also incentivizing positive transfer. I wonder how the authors would position their work relative to the body of work that optimizes for the gradient dot product. Why would we like gradients to be orthogonal if there would otherwise be transfer? Why focus on the cosine rather than the dot product, which naturally comes out of the first order Taylor expansion derivation for each task?

[4] "On First-Order Meta-Learning Algorithms". Alex Nichol, Joshua Achiam, John Schulman. 2018.

[5] "Learning to Learn without Forgetting by Maximizing Transfer and Minimizing Interference". Matthew Riemer, Ignacio Cases, Robert Ajemian, Miao Liu, Irina Rish, Yuhai Tu, and Gerald Tesauro. ICLR'19.

Given my major concerns about the theoretical motivation and comparisons to past work, I do not find the experiments comprehensive enough to prove the value of the proposed approach to the community. At the very least, I would be interested in comparison with additional very relevant baselines and in experiments with more tasks.

Update After Author Feedback:

While I really appreciate the authors providing some context about the references, I still feel like the paper would benefit from increased empirical comparison with these past approaches. Unfortunately, I don’t really follow the point they are making about why it is better to produce orthogonality vs. high dot products.  I agree that catastrophic forgetting is more of a problem related to continual learning, but I am not sure why we wouldn't want to maximize transfer even if there was no forgetting or interference.   Given my continued concerns, I am inclined to keep my score the same.

**Experience Assessment:**

I have published in this field for several years.

**Review Assessment: Checking Correctness Of Derivations And Theory:**

I assessed the sensibility of the derivations and theory.

**Review Assessment: Checking Correctness Of Experiments:**

I assessed the sensibility of the experiments.

**Review Assessment: Thoroughness In Paper Reading:**

I read the paper at least twice and used my best judgement in assessing the paper.

---

> ### Author Response · Authors · 2019-11-14
> **Response**
>
> We thank the reviewer for the feedback and for pointing out related work we have missed.
>
>
> 1. Cited papers related to orthogonal gradients.
>
> [1] focuses on the stability-plasticity dilemma in hyper-plastic noisy networks. The point they are making is that under conditions of large redundancy, extreme noise and high learning rates a multi-task network can still reach a dynamic equilibrium. They state that this is possible because at the point of convergence the solution will have orthogonal task gradients. As far as we can assess they don't make any reference to the orthogonality of task gradients during training. Furthermore, their theory is in line with our observations that in later stages of training the variance of the gradient distribution decreases as the cosine converges to 0. Our contribution however, lies in obtaining orthogonality throughout training which obtains competitive multi-task solutions.
>
> [2] propose a method for dynamically assigning parameters, or functional blocks, depending on the input and task. Their contribution is an architecture focused approach that is closely related to the soft parameter sharing paradigm. Our method on the other hand is a loss focused method suited exclusively for hard parameter sharing models. Consequently they achieve non interference by using different parameter partitions for different tasks, while we achieve this by regularizing gradients that affect the same set of parameters for all tasks. While both methods reduce task interference, the nature of the models they are designed for is entirely different.
>
> [3] devise a meta-learning algorithm that reduces task interference in a continual learning setting. Their method nudges learning of representations that are robust to interference, which turn out to be sparse. We agree that having sparse representations reduces task interference, and in a way it is related to our motivation behind CosReg - to force the optimizer to update parameters differently for each task during an iteration. Our method however doesn't directly induce sparsity constraints, which means it is less invasive on how the network should learn its representations. Future work can however investigate the learned representations.
>
> With respect to computational tractability - the execution time increases with the number of layers used for the gradient computation. Preliminary results seem to suggest however that CosReg is most effective on the layers closer to the encoders, which incur only a limited computational overhead. We will further quantify the performance when using gradients of different sizes and at varying positions.
>
> 2. Using the dot product as a penalty term.
> We believe there is a distinction to be made between training interference/transfer and overall performance in a MTL setting. Transfer is indeed maximized when gradients are pointing in the same direction but it is unclear whether this would also lead to a better performing solution in a multi-task setting. Standley et al. seem to make such a finding when analyzing what tasks can be learned together and our recent results on the SUN RGB-D dataset agree. Furthermore, the cited papers are in the domain of continual learning for which the main problem is catastrophic forgetting. In that setup it makes sense to have gradients of new tasks align with those of previous tasks. In a multi-task setting however there is no risk of forgetting as data from all tasks are available at all times.
>
>
> Refs:
> Which Tasks Should Be Learned Together in Multi-task Learning? Standley 2018

---

### Official Review · AnonReviewer3 · 2019-10-25
**Official Blind Review #3**

**Rating:** 3

**Review:**

The submission argues that when training the multiple objectives in a multi task learning framework, orthogonalizing gradients is beneficial toward reducing task competitions and efficient allocation of the learning capacity in the parameters.  This is demonstrated experimentally and subsequently a second order method is used for incorporating this in training.

Positives:
1. The concept is sensible, intuitive, and simple. There has a been quite few works (Sener and Koltun 2018) that similarly argue analyzing/regularizing the gradient direction when training multiple task objectives is beneficial. This submission reinforces those.

2. The implementation of the concept using a second order method is sensible and simple.

Weaknesses:
1. Missing comparisons: As reiterated above, several existing works have augmented training of neural networks with terms that are based on directions of gradients of multiple objectives. Some of them, e.g. Sener and Koltun 2018 or Du et al 2018, are very related. Though the submission cites them, no experimental comparison or convincing verbal critique of the differences are provided. The experiments could have included baselines that correspond to the concepts proposed in those prior papers to support the novelty of this submission. If the authors believe the concept of those papers are basically the same as theirs, then this submission should change to an analysis paper rather than appearing to pitch a new regularization term. I would have found an analysis paper as valuable as one with a novel method, but the stance should be clear.

2. I found the experiments too toy to be convincing. The MultiDigitMNIST is artificial and with limited benchmarking value as its construct doesn't really reflect the construct of multi task learning in the real world (e.g. as in the multi task vision datasets). NYUv2 dataset is more realistic, but the reported results are not clear to show significant and meaningful differences (see table 2). Particularly since the NYUv2 dataset has certain biases with imperfect ground truth from the kinect sensor which questions if differences in 0.001 range are meaningful. The authors can consider more recent and comparatively more reliable multi label datasets like Taskonomy for benchmarking.

3. Inline with the above comment, qualitative results or any other method for convincing that the achieved results (on datasets other than MultiDigitMNIST) is meaningful seems crucial.

Further comments:
4. The single task baselines in table 2 often perform inferior to the multi task baselines. This suggests the NYUv2 dataset could be too small to learn individual networks, whereas many other multi task papers often find single task baselines are hard to beat if they're not starved of parameters (e.g. see Standley 2018 "Which Tasks Should Be Learned Together in Multi-task Learning?"). Please clarify and/or consider using large enough datasets that allow you to benchmark under both high data and low data regimes.

5. why there is no single task baseline in table 1?

6. why CosReg is not included in figure 3 plot?

7. A closer look at the recent works on analyzing task competitions in multi task learning could be useful, and probably supportive of the concept of this submission. For instance standley 2018 Which Tasks Should Be Learned Together in Multi-task Learning seem to suggest that tasks that are related under transfer learning setting did not help each other in multi task setting. Their observation seems related to the pitch of this paper that orthogonal gradients (ie tasks with dissimilar updates) can be optimized better.

8. In page 4 authors state that computing the regularization term for all layers is complex, hence they do that only for the last layer. This seems okay to me, though I would have found an experiment demonstrating the consequences of this simplification useful (e.g. am experiment showing which layer to pick and a one-time expensive experiment demonstrating that picking one layer vs more layers is not too damaging).

-------
Comments after rebuttal:
I read the rebuttal and appreciate authors responses and the attempted improvement. It helped. I think the submission has the potential to be ultimately a useful read for the community, but at this stage more work/revision that wouldn't fit rebuttal time constraints would be needed to achieve that. I still think an experimental comparison with the related methods would be more convincing than verbal.

The added experiment on SUN RGB-D dataset is appreciated, but all results appear too close (table 2) to suggest one should adopt the proposed method. Also SUN RGB-D limits the tasks that could be learned together in a multi-task experiment given the limited number of labels per datapoint. I would have found using more recent  datasets with higher number of labels (rather than forcing to use depth+semantics which doesn't have to be a good mix, see Standley 2018) and more significant quantitate improvements more convincing.

Overall I lean toward not accepting the current submission but acknowledge the potential value in resubmission after further work.


**Experience Assessment:**

I have published in this field for several years.

**Review Assessment: Checking Correctness Of Derivations And Theory:**

I carefully checked the derivations and theory.

**Review Assessment: Checking Correctness Of Experiments:**

I carefully checked the experiments.

**Review Assessment: Thoroughness In Paper Reading:**

I read the paper thoroughly.

---

> ### Author Response · Authors · 2019-11-14
> **Response**
>
> We thank the reviewer for the detailed and constructive feedback, giving us a useful direction for improving our paper.
>
>
> 1. Comparison with Sener et al.
> Sener et al. also take a close look at the task gradients of the shared parameters but with the goal to then rescale these such that their convex combination will satisfy the necessary conditions to reach a Pareto optimal solution. In that sense they are closer aligned to Gradnorm and Kendall, who find task weights that ultimately resize the gradient magnitudes. Our method forces a different optimization trajectory by favoring specific locations on the loss surface which have orthogonal task gradients, without using task weights. From a point of view of the resulting gradient a similarity we share is that non-interference is assured at each update. For us this is done by having orthogonal task gradients, for Sener et al. it is achieved by using MGDA updates, which guarantee that the direction of the final gradient improves all tasks. We have added this differentiation to our paper, but unfortunately didn't have enough time to conduct benchmark experiments.
>
> 2 & 3. Experiments on more/larger datasets
> We have added experiments on the SUN RGB-D dataset, which is not as large as Taskonomy but still offers an order of magnitude more data than NYUv2. These new experiments allow us to test our method when there is a strong relatedness/transfer between tasks - fine and coarse semantic segmentation. We have also added a figure showing the cosine distribution during training and how our method successfully concentrates it around 0. This further strengthens our argument since we manage to outperform the benchmarks for which task gradients point in a similar direction.
>
> 4 & 5. Low/High data regimes and single task benchmarks
> We believe the inferior performance of the single task baselines is due to the complexity of the dataset relative to the amount of available data, as we have observed a similar pattern on the SUN RGB-D dataset. Regarding the missing single task baselines for the multiDigitMNIST dataset, we initially just wanted to emphasize the effect of regularization in a multi-task setting but agree that including these baselines shows the bigger picture. In this case the single task baselines outperform the multi-task approaches, which agrees with our hypothesis since the dataset offers a large amount of data compared to its difficulty.
>
> 6. We have not included CosReg in figure 3 as we wanted to show the naturally occurring relationship between cosine standard deviation and final performance of different regularization methods. For CosReg there would be close to no variation in standard deviation since the method seeks to minimize it.
>
> 7. Task relatedness and MTL performance
> We agree and thank the reviewer for pointing this out. Our recent experiments with related segmentation tasks on the SUN RGB-D dataset seem enforce this view.
>
> 8. CosReg ablation
> We agree about the usefulness of such experiments, but unfortunately weren't able to complete a comprehensive set of experiments on time. Preliminary results seem to indicate that it is best to place CosReg on the layers closest to the encoders.

---

### Decision · Program_Chairs · 2019-12-19

**Decision:**

Reject

**Comment:**

This paper proposes a training approach that orthogonalizes gradients to enable better learning across multiple tasks. The idea is simple and intuitive.

Given that there is past work following the same kind of ideas, it would be need to further:
(a) expand the experimental evaluation section with comparisons to prior work and, ideally, demonstrate stronger results.
 (b) study in more depth the assumptions behind gradient orthogonality for transfer. This would increase impact on top of past literature by explaining, besides intuitions, why gradient orthogonality helps for transfer in the first place.